# Toward Wearable MagnetoCardioGraphy (MCG) for Cognitive Workload Monitoring: Advancements in Sensor and Study Design

**DOI:** 10.3390/s25154806

**Published:** 2025-08-05

**Authors:** Ali Kaiss, Jingzhen Yang, Asimina Kiourti

**Affiliations:** 1Department of Electrical and Computer Engineering, The Ohio State University, Columbus, OH 43210, USA; kiourti.1@osu.edu; 2Center for Research Injury Research and Policy, Abigail Wexner Research Institute at Nationwide Children’s Hospital, Columbus, OH 43205, USA; ginger.yang@nationwidechildrens.org

**Keywords:** cognitive workload (CW), ElectroCardioGraphy (ECG), heart rate variability (HRV), MagnetoCardioGraphy (MCG)

## Abstract

Despite cognitive workload (CW) being a critical metric in several applications, no technology exists to seamlessly and reliably quantify CW. Previously, we demonstrated the feasibility of a wearable MagnetoCardioGraphy (MCG) sensor to classify high vs. low CW based on MCG-derived heart rate variability (mHRV). However, our sensor was unable to address certain critical operational requirements, resulting in noisy signals, often to the point of being unusable. In addition, test conditions for the participants were not decoupled from motion (i.e., physical activity (PA)), raising questions as to whether the noted changes in mHRV were attributed to CW, PA, or both. This study reports software and hardware advancements to optimize the MCG data quality, and investigates whether changes in CW (in the absence of PA) can be reliably detected. Performance is validated for healthy adults (n = 10) performing three types of CW tasks (one for low CW and two for high CW to eliminate the memory effect). Results demonstrate the ability to retrieve MCG R-peaks throughout the recordings, as well as the ability to differentiate high vs. low CW in all cases, confirming that CW does modulate the mHRV. A paired Bonferroni t-test with significance α=0.01 confirms the hypothesis that an increase in CW decreases mHRV. Our findings lay the groundwork toward a seamless, practical, and low-cost sensor for monitoring CW.

## 1. Introduction

Cognitive workload (CW) is defined as the mental effort exerted during a mental task [1,2]. Regulating CW is widely recognized as a means of enhancing human performance, reducing the likelihood of errors, and supporting the optimal physical and mental well-being of individuals [3,4]. Monitoring CW is especially crucial in safety-critical settings like cockpits, air traffic control, and driving, where even minor errors can have severe consequences [5]. Similarly, CW assessment is critical in a variety of clinical and healthcare contexts, where mental effort, attention, or fatigue can significantly affect safety, performance, and therapeutic outcomes [6,7,8].

Today’s gold-standard for monitoring CW relies on self-reporting methods that are subjective/biased and, hence, unreliable [9]. Objective physiological measures have been explored for assessing CW. ElectroEncephaloGraphy (EEG), MagnetoEncephaloGraphy (MEG), functional magnetic resonance imaging (fMRI), positron emission tomography (PET), and eye tracking/pupillometry are the most common physiological measures of CW. EEG directly captures the brain’s electrical activity through electrodes positioned on the scalp. This method tracks variations in CW over time and delivers results with high temporal precision [10]. MEG measures the magnetic fields generated by the brain’s electrical activity. It has shown to achieve similar accuracies as EEG [11]. fMRI detects changes in blood flow related to neural activity. It measures the signal that is dependent on blood oxygenation level and has shown to monitor brain activity with high spatial resolution [12,13]. PET observes metabolic processes in the brain. It can monitor dopamine receptors at rest and while performing a cognitive task [14]. Task-evoked pupillary responses (TEPR) have also shown to effectively estimate the cognitive effort involved in completing a task [15]. However, although the aforementioned techniques have shown to be effective, they can be very complicated to construct and operate or very expensive to obtain.

To address such challenges, we recently introduced a wearable MagnetoCardioGraphy (MCG) sensor with the ability to differentiate high vs. low CW using MCG-derived heart rate variability (mHRV) metrics [16]. Here, we refer to the MCG-derived HRV as mHRV to distinguish it from ElectroCardioGraphy (ECG)-based HRV, noting that while MCG and ECG originate from the same source, they are not identical (much as pulse rate variability, or PRV, is not identical to HRV [17]). As is well known, a healthy heart does not beat at a constant rhythm; instead, its natural fluctuations enable the cardiovascular system to quickly respond to sudden physical or mental demands that disrupt the body’s balance. Accordingly, the autonomic nervous system regulates heart function in response to CW: high CW tends to increase sympathetic activity and suppress parasympathetic activity, leading to reduced beat-to-beat variability. To this end, numerous studies have utilized HRV as an indirect measure of CW [18,19,20,21,22]. However, unlike ECG, MCG has numerous advantages, including a non-contact operation, insensitivity to hair and sweat, and robustness to changes in biological tissue properties (since tissues are non-magnetic) [23]. Notably, we have been the first to explore the classification of CW using mHRV acquired from wearable low-cost MCG sensors [16]. In brief, our first such sensors consisted of an array of seven (7) miniature coils placed upon the chest to passively capture the naturally emanated cardiac magnetic fields. To visualize such signals, the raw data collected by the sensor were amplified and post-processed, as detailed in [16].

Despite the promising results, our study [16]—as the first of its kind—suffered from several limitations. First, the MCG sensor hardware exhibited structural fragility, making it susceptible to frequent malfunctions and noisy data collection. Specifically, the outputs of each of the sensing coils were soldered to jumper wires that were, in turn, soldered to an amplifier board. The presence of these 14 wires (2 for each coil, with a total of 7 coils) not only limited the mobility of the sensor but also increased the system noise. Second, the coils were not embedded in appropriate mounting fixtures, causing them to jitter, further contributing to noise. Third, the MCG sensor was secured to the subject’s chest with gauze tape, causing unwanted vibrations during use and further increasing the noise. Fourth, the employed signal processing (simply averaging across the coils and filtering) lacked the sophistication required to de-noise the data. As a result, the MCG data from some subjects had to be completely discarded, while even the `good’ recordings had to be reduced to 0.6 times the original duration (not necessarily continuous in time) to ensure that the signal R-peaks were visible. Finally, the test conditions used for the participants were not decoupled from motion (i.e., physical activity, PA), raising concerns as to whether the noted changes in mHRV were attributed to CW, PA, or both. Specifically, participants were recording their answers for the high CW task manually on a cell phone, whereas the literature (and our own studies) have shown that even subtle movements or stress responses can influence the mHRV [16,24,25].

In a major step forward, we herewith introduce significant advancements in hardware architecture, signal processing methodologies, and study design to overcome the above-mentioned limitations. The approach was validated for (n = 10) healthy adult participants performing three types of CW tasks (one for low CW and two for high CW to eliminate the memory effect). Our work resulted in a robust and accurate MCG sensor with wearable form-factor and low-cost fabrication, demonstrated to retrieve the MCG R-peaks for the entire duration of the recordings and classify high vs. low CW in 100% of the cases. Our findings also confirm that CW alone does modulate mHRV, regardless of the presence of PA and, thus, lays the groundwork for future work in the area of MCG-based CW classification. In summary, the key advantage of our work is leveraging MCG as a completely passive cardiac measure for CW. Unlike previous technologies for CW monitoring, our approach requires no skin contact; it senses the heart’s magnetic field directly. This enables continuous monitoring without electrode setup or skin irritation. Furthermore, we operate in normal ambient conditions (no shielding), which is novel. In addition, the setup is very cheap to build and operate, as opposed to PET and MEG. Aside from the hardware aspects, another key contribution of this work lies in the separation of changes in HRV from CW and PA. Although the literature has confirmed the relationship between changes in CW and HRV, no published work exists that tracks the changes back to CW.

The rest of the paper is organized as follows: Section 2 provides information on the MCG sensor hardware, signal processing methodology, study scenarios, and study participants. Experimental results are reported in Section 3. We discuss the results, practical/clinical implications of our findings, and future work in Section 4. The paper concludes in Section 5.

## 2. Materials and Methods

### 2.1. MCG Sensor Hardware

The MCG sensor employed in this work is shown in Figure 1a and was intended to capture the naturally emanated magnetic fields of the heart, namely the MCG signal. One potential concern that may arise with the wearable MCG sensor is the possibility of the resulting signal being attributed to heart vibrations or sounds as opposed to MCG. However, as addressed in our previous research [26], the recorded signal is solely attributed to the heart’s magnetic field, and the sensor does not measure acoustic heart sounds. Indeed, the coil design has been specifically optimized to pick up MCG signals, while the accompanying signal processing further helps suppress noise while preserving the cardiac signal. The sensor consisted of an array of eight (8) coils (each 12 mm in height and 16.6 mm in outer diameter), embedded within a 3D-printed fixture of 95 mm in diameter. The coils were the same as those used in [16]; however, their total number was increased from 7 (in [16]) to 8 (in this work), to enhance uncorrelated noise suppression. Specifically, averaging the same MCG signal over *N* number of coils is expected to reduce the noise by $\sqrt{N}$. The number *N* = 8 was optimized for a maximum number of coils fitting within the 95 mm diameter of the sensor, with the latter diameter selected for an optimal fit upon the average human heart.

The initial optimization parameters for the coil design were previously established in [27], where the induction coil sensor was designed based on the model of a tightly wound air core coil with an inner diameter $D_{i}$, outer diameter *D*, length *L*, and wire diameter *d*. In this case, the sensitivity can be written as(1)$$\frac{S}{D^{2.5}}=M\left(1+\frac{D}{D_{i}}\right){\left[\frac{1-{\left(\frac{D_{i}}{D}\right)}^{5}}{1-{\left(\frac{D_{i}}{D}\right)}^{3}}\right]}^{\frac{1}{4}}\sqrt{(1-{\left(\frac{D_{i}}{D}\right)}^{2}}$$
and obtains maximum values if $\frac{D_{i}}{D}\approx 0.6$ and $\frac{L}{D}\approx 0.7$. Therefore, by selecting a value for *D*, optimal values for $D_{i}$ and *L* can be chosen. As for the array holding these coils, the design specifications were constrained by two main criteria:An array that can properly hold the coils in place,A sensor that can be properly fixed on the chest with no jittering.

To address the first criterion, we 3D-printed a design with holes that are equivalent to the abovementioned length, *L*. By doing this, we made sure that the coils are held in place without moving. To address the second criterion, we added hooks and introduced ratchet straps so that the sensor will remain in place.

Contrary to [16] where the coils where simply positioned on a plastic base and secured on this base, as well as upon the wearer, with tape (see Figure 1b), a new 3D-printed fixture with a 3-layer structure was introduced. The following explains in details the different layers of the sensor as shown in Figure 2:The bottom layer included a dedicated pocket (each 12 mm deep) to keep each of the coils in place and avoid jittering. This step is critical, as jittering can misalign the waveforms of each of the 8 coils: any misalignment between heart beats results in imperfect constructive averaging, reducing the effectiveness of noise cancellation. In fact, even small temporal jitters can blur sharp features (e.g., QRS complexes), limiting the signal-to-noise ratio (SNR) gain. The bottom layer also included hooks that accommodated 3 ratchet straps to hold the sensor in place on the subject’s chest (see Figure 3). There were 8 hooks in total with 6 serving to hold the ratchet straps attachments (2 hooks per strap) and 2 serving as latches for the top layer of the sensor. At the end of each ratchet strap, we placed a tightening buckle as needed to better adjust the size and enhance the participant’s comfort (see Figure 3).The middle layer served as a platform that held the top layer (namely the array connector in Figure 2) on top of the coils. Inside this layer were small holes that allowed the output of each of the coils (i.e., two wires for every coil), to be soldered to the base of the array connector. In other words, the input to the array connector was the output of the coils. In doing this, the wires were properly organized within the sensor to avoid tangling.The top layer (namely, the array connector) served to convert the outputs of the 8 coils into a two-ethernet cable signal, as shown in Figure 1a. This was a major improvement compared to Figure 1b where the coil outputs were coming directly out of the sensor in a tangled way, increasing the noise.

The structure was 3D-printed using a Bambu Lab X1-Carbon 3D-printer (Bambu Lab, Shenzhen, China). We selected Polyethylene Terephthalate Glycol-modified (PETG) material that offered a balance between ease of printing and mechanical strength. Notably, PETG is particularly suitable for functional parts that require durability, impact resistance, and water resistance. The material is non-magnetic and, hence, transparent to the collected MCG signals.

The complete MCG sensor setup is shown in Figure 4. Specifically, the two ethernet cables from the output of the array connector were connected to an amplifier board to improve the noise figure of the system and make it more robust. Similarly to [27], the amplifiers incorporated an input network to suppress oscillations in the input port, were placed a distance away from the sensor coils to reduce noise, and were all mounted on a single board to reduce relative motion/vibrations. Once the MCG signals from each of the coils were amplified, they passed through a 24-bit multi-channel Analog-to-Digital Converter (ADC) from National Instruments (Austin, TX, USA). The ADC operated with a ±10 V power supply and was configured to sample at 5 kHz. The digitized signals were then set as input to a laptop computer for post-processing, as discussed next.

### 2.2. MCG Signal Processing

Referring to Figure 5, the first step in signal processing was to filter each of the 8 MCG signals and remove high-frequency noise, such as electrical interference. To do so, a digital band-pass filter in the range [6–36] Hz was applied. The second step was to apply averaging across the 8 band-pass filtered signals. The reasoning behind this averaging was discussed in Section 2.1. To further de-noise the signal and detect the location of each of the heart beats (as needed to retrieve the mHRV), we applied our recently reported algorithm, beat estimation [28]. beat estimation leverages signal averaging and template matching to robustly identify heart beats from noisy MCG signals, achieving a dramatic improvement in the R-peak detection accuracy over the state of the art and nearly perfect mHRV estimation. As a last step, the mHRV was computed. Here, we calculated the mHRV as the mean of the difference in duration between R-peaks, denoted as $Mean_{RR}$, which was calculated according to(2)$$Mean_{RR}=\frac{1}{N-1}\sum_{i=1}^{N-1}R_{i+1}-R_{i},$$
where $R_{i}$ is the index of an R-peak obtained through beat estimation. Other metrics, like the standard deviation (SD), were not calculated because over a short period of time (e.g., 5 to 10 min of recordings considered in this study), the duration between heart beats is constantly changing: while this does not significantly affect the mean, it does render using a constant mean model to calculate SD invalid. Therefore, to calculate the SD, assumptions on the variations of the mean over a short period of time should be taken, and then, from the derived model for the time-varying mean, a time-varying SD can be calculated.

### 2.3. Study Design

Human subjects were enrolled to participate in three (3) different scenarios of exerted CW. These scenarios were designed to validate the following hypotheses: (a) the MCG sensor setup reported in this study can classify high vs. low CW, validating the results of [16] for a different hardware and algorithmic setup, and (b) CW modulates the mHRV, even in the absence of any PA. Specifically, the testing scenarios were (see Figure 6) as follows:1.Scenario 1: Low CW: The subject was sitting on an office chair while watching a relaxing video. The subject was asked to refrain from speaking and performing any type of motion. The testing duration was 7.5 min.2.Scenario 2: High CW with PA: The subject was sitting on an office chair, performing N-back tasks while also recording their answer (True or False) on a phone held on their dominant hand. For this experiment, the subject only answered `true’ when the current stimulus matched the stimulus from 2 steps earlier. To increase the level of difficulty, a mix of numbers (1 and 2), letters (A and B), and shapes (triangle and circle) were used as the stimulus. The subject was asked to refrain from speaking, but motion was allowed. The testing duration was 7.5 min.3.Scenario 3: High CW without PA: The subject was sitting on an office chair while mentally answering `true’ or `false’ for two-digit addition and subtraction math equations. The subject was asked to refrain from speaking and performing any type of motion. The testing duration was 7.5 min.

Our first hypothesis was to reproduce the results in [16] for the new sensor (hardware and signal processing) and relied on Scenario 1 and Scenario 2. Our second hypothesis was to clarify the uncertainty raised in [16] as to whether the mHRV changes are due to CW, PA, or both and relied on Scenario 1 and Scenario 3. The high CW tasks in Scenario 2 and 3 were intentionally selected as different to eliminate the memory effect. The selection of the tasks themselves relied on previous studies for the exertion of low and high CW [24,25].

For all scenarios, the following sensors were placed on the human subjects and used to record data simultaneously (see Figure 7):

MCG sensor: The MCG sensor was described in Section 2.1 and Section 2.2. For proper placement on the chest, we counted from the clavicle and down to the space between the third and fourth ribs to identify the location of the heart and aligned the MCG sensor with this location.ECG sensor: A three-lead off-the-shelf Arduino UNO R3 micro-controller board (Arduino S.r.l, Ivrea, Italy) was used. The ECG signal served as a `gold standard’ comparison vs. the results obtained through our MCG sensor. The ECG sensor output was connected to one of the ADC channels, and the signal processing followed the steps in Figure 5 (except the averaging).Inertial Measurement Unit (IMU): A Witmotion WT9011DCL MPU9250 Bluetooth accelerometer (WitMotion Shenzhen Co., Ltd., Shenzhen, China) was placed on the opposite side of the palm of the subject’s dominant hand, i.e., the one used to hold the phone in Scenario 2. The sensor had dimensions of 32.5 mm × 23.5 mm × 11.4 mm and was used to monitor the presence/lack of PA.Finger Pulse Oximeter: A fingertip pulse oximeter SM-1100S (Gurin Products, LLC, Tustin, CA, USA) was used to measure oxygen saturation. Although data from the oxygen sensor was not used in the post-processing, it was utilized as assurance that the participants were not in distress.

### 2.4. Study Participants

For this study, 10 adults between the ages of 19 and 33 (μ = 23.5 years; σ = 4.11 years) were recruited, as shown in Table 1. For this proof-of-concept study, participants were selected as healthy (i.e., no cardiac conditions) to ensure that changes in mHRV were specifically attributed to CW. Our inclusion criteria (approved by the Institutional Review Board (IRB)) required “healthy” status and normal BMI. No screening for subclinical conditions (like undiagnosed arrhythmias) was performed, but none of the volunteers reported any cardiovascular disease or took cardiac medications. This is important because pre-existing cardiac conditions can significantly affect the HRV. For example, pathological arrhythmias or heart disease often reduce HRV and alter its pattern [29]. By restricting to healthy adults, we ensured our HRV changes reflected cognitive load rather than underlying pathology. For future studies involving participants with known or suspected cardiac conditions, we can leverage the partnership with Nationwide Children’s Hospital to recruit patients who have already been appropriately screened. This approach helps ensure that our findings are not confounded by underlying cardiac issues. Additionally, we can use a three-lead ECG system to aid in detecting arrhythmias, as different types of cardiac irregularities are known to alter the morphology of the heartbeat [30,31]. To determine the sample size of human subjects for validation, we based our decision on previous studies of CW monitoring, which included 10 [32], 16 [33], 12 [34], 9 [35,36], and 11 [29] participants. All participants were males due to the garment being more comfortable in the absence of breast tissue (note the tightening of the ratchet straps in the chest area in Figure 3). The protocol was approved by The Ohio State University Institutional Review Board (protocol # 2019H0259). Though exact measurements of the heart-to-coil distance were not possible to acquire, all subjects had a healthy body mass index (BMI < 30 kg/m^2^).

## 3. Results

### 3.1. Efficacy of the MCG Sensor Hardware and Signal Processing Advancements

To emphasize the improvements that resulted from the advancements in the MCG sensor hardware, Figure 8a shows a zoom-in on an example MCG signal after band-pass filtering and averaging, but before applying beat estimation. For comparison, Figure 8b shows the equivalent signal using the hardware setup in [16]. `Gold standard’ ECG signals are also super-imposed with a goal to pinpoint the accurate retrieval of the R-peaks. As seen, the signal in Figure 8a is much clearer, and the R-peaks can be seen even in the absence of advanced signal processing. To quantify the performance improvement, we calculated the total number of detected R-peaks in the MCG signal and divided that by the total number of detected R-peaks in the ECG signal (i.e., the ground truth) as shown in Equation (Equation 3). For the full recording (duration = 3 min) in Figure 8b, the detection accuracy was only 70.92%, whereas for the one in Figure 8a (duration = 7.5 min), it was a perfect 100%. That is, the hardware advancements proposed in this paper drastically improve the detection of the R-peaks.(3)$$\mathrm{Detection}\;\mathrm{Accuracy}=\frac{\mathrm{\# of detected R-peaks}}{\mathrm{\# of true R-peaks}}\times 100$$

When beat estimation is also taken into account, the performance improvement of the proposed MCG sensor is highlighted, as shown in Figure 9a. A clear and accurate detection of the R-peaks is taking place. Although this is also true for Figure 9b that shows the corresponding data for the sensor in [16], the noise level is much lower in Figure 9a. Table 2 provides a summary of the detection accuracy of all subjects in each of the scenarios after applying beat estimation. Scenario 3 had the highest average accuracy of 99.3%, Scenario 2 had the lowest average accuracy of 98.2 %, and Scenario 1 had an average accuracy of 98.4 %. This shows that our obtained R-peak data are accurate, valid, and comparable to those obtained from the ECG sensor.

### 3.2. Efficacy of the Testing Scenarios

For Scenario 1, the subjects were monitored throughout the duration of the testing to ensure that they were fully engaged with the relaxing video being played in front of them. For Scenario 2, we calculated the subjects’ accuracy in performing the N-back tasks using their answers recorded on the phone. All 10 subjects achieved an accuracy of over 80%. This confirms that the intended CW was actually exerted. For Scenario 3, since no physical recording of the answers was available, the subjects verbally confirmed their engagement in the task once the recording was over.

As for the PA engagement, Figure 10a–c show the IMU recordings in both X and Y directions in units of gravitational acceleration g (m/s^2^) for Scenarios 1 through 3 for one of the subjects. The results confirm that there was minimal to no motion in Scenarios 1 and 3, unlike Scenario 2 where motion was recorded due to the subjects noting the answers of the N-back tasks on the phone. Indeed, Figure 10b shows multiple spikes throughout the presented duration of 20 s, whereas the other Figures do not. Accordingly, the variance was very low in both Figure 10a,c, unlike that of Scenario 2. This behavior was observed across all 10 subjects.

### 3.3. HRV Results

Figure 11 plots the maximum, mean, and minimum values of the $Mean_{RR}$ obtained through MCG and ECG for each of the testing scenarios. To obtain this plot, we considered the average over the 10 subjects for each scenario. By calculating the percentage error for the mean between the ECG and MCG, we found out that a maximum error of 0.16% occurred in Scenario 1, the lowest error of 0.02% occurred for Scenario 3, while Scenario 2 had an error of 0.15%. As for the maximum, the highest error of 0.55% occurred in Scenario 2, the lowest error of 0.27% occurred in Scenario 3, while Scenario 1 achieved an error of 0.48%. Finally, the error values for the minimum ranged from 0.19% as the highest error in Scenario 3, to almost 0% in Scenario 1, while Scenario 2 had an error of only 0.16%. That is, our mHRV results are almost the same results as those obtained from ECG, which has a very high SNR and is known as the ground truth, with errors not exceeding 0.6% on average when it comes to the minimum, maximum, and mean values of calculated $Mean_{RR}$. These results further confirm the adequacy of our hardware and algorithmic setup.

To prove our hypotheses, Figure 12a,c show box plots for the MCG for Scenario 1 vs. Scenario 2 and Scenario 1 vs. Scenario 3, respectively. For comparison and validation against the `gold standard’, Figure 12b,d represent the equivalent box plots for the ECG. In all figures, the red box represents the $Mean_{RR}$ calculated for Scenario 1, with a red circle indicating the median of all participants in that scenario. The green box in Figure 12a,b represents the $Mean_{RR}$ calculated for Scenario 2, with a green circle indicating the median of all participants in that scenario. Lastly, the green box in Figure 12c,d represents the $Mean_{RR}$ calculated for Scenario 3, with a blue circle indicating the median of all participants in that scenario. The blue line connecting all boxes together is the mean of each scenario taken across all participants.

Figure 12a,b show that the mHRV and HRV drop from baseline when both CW and PA activity are exerted. This part of the experiment was conducted to compare to the one in [16] and confirm the results. However, in [16] and in Figure 12a,b, it is not evident whether the drop occurs due to CW, PA, or both. To confirm that the changes in CW indeed cause drops in the mHRV and HRV, we compared Scenario 1 to Scenario 3 (noting the difference in the high CW task as compared to Scenario 2 to avoid the memory effect). Figure 12c,d show the obtained results. Since only CW changes occur between these two scenarios, as also confirmed by the IMU readings, we validate our hypothesis that the mHRV and HRV drop with increasing CW levels, regardless of the presence of PA.

To confirm the significance of HRV differences across scenarios, we performed a paired Bonferroni *t*-test with significance α=0.01 that confirmed our hypothesis. Let $\mu_{i}$ denote the sample mean of the *i*th scenario for i∈{1,2,3}. We obtained a *p*-value of $p^{\left(1\right)}=2.2\times 10^{-3}$ for $H_{0}^{\left(1\right)}:\mu_{1}<\mu_{2}$ and a *p*-value of $p^{\left(2\right)}=4.28\times 10^{-4}$ for $H_{0}^{\left(2\right)}:\mu_{1}<\mu_{3}$.

## 4. Discussion

In this study, we observed a clear decrease in mHRV during periods of high CW compared to a low CW baseline, even when the participants remained motionless. Indeed, tasks that imposed greater mental demands—such as an N-back working memory challenge or complex arithmetic problem solving—elicited significantly lower mHRV values relative to the resting baseline condition (in our case, watching a relaxing video). This drop in mHRV with increasing CW, despite the absence of any PA, indicates that the autonomic nervous system shifts toward sympathetic dominance (and/or vagal withdrawal) purely due to mental effort [37]. Our findings are the first to decouple the effects of mental workload on mHRV from any physical influences. This interpretation aligns with established physiological responses: mental strain is known to suppress the high-frequency (parasympathetic) component of HRV [38], leading to an overall reduction in variability. Our results bolster the evidence that mHRV is a sensitive marker of cognitive strain. Furthermore, we observed this mHRV decline consistently across different types of cognitive tasks, suggesting a robust autonomic response to increased mental effort.

Notably, the literature suggests that CW can alter HRV even when the mean heart rate or other physiological measures (such as blood pressure) remain relatively unchanged [37,39]. This underscores the unique benefits of our approach: continuous monitoring of mHRV can capture subtle shifts in autonomic balance that might be missed by looking only at average heart rate or other coarse vital signs [38]. Importantly, the MCG sensor sidesteps some challenges faced by ECG-based methods. Because MCG detects magnetic fields, it is immune to issues like electrode placement variability, skin impedance changes (due to sweat or motion), and the need for skin contact, conductive gels, or adhesive patches. This can translate to more seamless operation and improved comfort for day-to-day use of the sensor.

When comparing our results to prior ECG- and MCG-based studies of CW, we find strong agreement. In laboratory tasks such as the *N*-back working memory challenge, others have reported pronounced decreases in time-domain ECG-based HRV indices as the task difficulty increases [40]. The authors concluded that HRV metrics provide a valid index of cognitive load, correlating with subjective workload and driver performance measures. In our own previous work [16], we demonstrated that a wearable non-contact MCG sensor could distinguish high vs. low CW by analyzing standard mHRV metrics. However, such prior ECG and MCG works still involve minimal movements, leaving some room for doubt about subtle physical influences. Our findings in a controlled motionless setting complement such prior studies by confirming that the reduction in mHRV is truly due to CW itself. This adds credence to the idea that HRV could serve as a general-purpose indicator of cognitive strain across a spectrum of contexts—from a person quietly solving math problems (as in our study) to a multitasking driver in motion—as long as one accounts for or isolates the confounding influences.

The potential for a future wearable implementation of the system is confirmed by our ability to identify inter-beat intervals even when the raw magnetic signal was barely distinguishable by eye. In essence, the hardware design and accompanying signal processing (particularly beat estimation) boost the effective SNR by aggregating information over time and intelligently rejecting noise, thus enabling accurate mHRV computation where a naive approach would fail. The reported technical enhancements in this work greatly improve the signal reliability and confidence in using MCG recorded from wearable sensors for HRV analysis. By demonstrating that we can derive stable mHRV measurements from such a challenging signal environment, we have mitigated one of the primary concerns regarding wearable MCG deployment. Previous work had suggested the promise of using un-cooled portable magnetometer sensors for bio-signal capture, but issues of noise and motion artifacts were major barriers [16,27]. In our implementation, not only did the beat estimation handle intrinsic sensor noise, it also dealt effectively with any minor disturbances (such as environmental electromagnetic fluctuations or the slight shifts in sensor position relative to the heart).

As is expected, CW can manifest in many contexts beyond the scenarios explored in this pilot study. The selection of the subject scenarios relied on paradigms that have been widely used in the literature to reliably modulate the level of mental effort [32], while the choice of having a controlled laboratory setting arises from the fact that we want to track changes in mHRV solely due to CW. Other memory/cognitive tasks in controlled environments are expected to yield similar results: even with different intensities, mHRV is expected to drop whenever CW increases. Beyond these scenarios, our MCG approach should, in principle, apply to any task that triggers an autonomic response. For instance, demanding cognitive tasks in safety-critical domains, such as air-traffic control or surgical training, have been shown to tax cognitive resources and mHRV [41]. Likewise, tasks in healthcare (e.g., concussion recovery exercises), education (exam-taking, tutoring), gaming, and immersive training could all benefit from workload monitoring [42,43]. However, applying the proposed approach directly in real-world scenarios would introduce extra variables that might impact the mHRV, i.e., mHRV is influenced by several factors beyond CW, such as stress or fitness levels [44]. As such, for this proof-of-concept study, the experimental scenarios were carefully chosen to mimic real-life activities while at the same time limiting external variables that might hinder the mHRV. By contrast, in uncontrolled environments, cognitive load changes might be confounded by other stressors. Any real-world system must account for these. In summary, even though the proposed MCG sensor can be used wherever continuous unobtrusive monitoring of CW is needed, such as applications that include safety monitoring (e.g., warning drowsy or cognitively overloaded drivers/pilots), workplace optimization (e.g., measuring staff workload in manufacturing or control-room environments), and consumer devices (e.g., games or virtual reality systems that adapt to user effort), it is still in the testing stage. Optimizing the sensor for daily life activities presents several challenges, particularly on the hardware front. While the current amplifier is portable, it is not yet small enough to be considered wearable. Further development is needed to refine and downsize the amplifier so it can match the dimensions of the MCG sensor and be mounted directly on top. Additionally, the ADC, which digitizes the recorded signals for further processing, should ideally be integrated into the same board as the amplifier. Future research should explore the use of an external electromagnetic transducer designed to suppress common-mode noise. On the algorithmic side, while techniques like adaptive filtering [45], which utilizes data from an IMU, and Independent Component Analysis (ICA) [46] are being considered, incorporating an IMU alone is not sufficient. It is equally important to understand the origins of these motion artifacts in order to model and effectively eliminate them.

Though females were not included in this study, we expect the results obtained in this study to be generalizable. That is, a similar response in mHRV is anticipated with increasing CW, regardless of the participant’s sex. Specifically, the basic autonomic mechanisms (i.e., how CW affects HRV) are qualitatively similar across sexes, even if baseline values differ. Known sex differences do exist (for example, women tend to have a slightly higher resting heart rate), but no evidence suggests opposite directions of the CW effect. This was shown in several similar works that address CW issues and used only males [47,48], or only females [49] in their testing. This was also demonstrated in our previous work [16], where it was proven that increasing CW will lead to a drop in HRV, among both males and females. Thus, although baseline variations might occur, i.e., variations in the significance of the drop in HRV, HRV is still expected to drop whenever CW increases in both males and females. As for concerns regarding the sensitivity of the sensor due to variations in breast tissue size between males and females, this was proved to have little to not effect during our experiments, as our subjects had varying chest sizes, especially since most of them work out. In our future work, we will aim to modify the garment to make it suitable for female participants as well. This can be achieved by improving the bottom layer of the sensor, i.e., the layer in direct contact with the chest, to a new material that can adjust with different chest formations and relying on stretchable wires to hold the sensor in place. To make our results more statistically valid, a larger pool of subjects, including children, will be tested on in the future. Though the participant sample size is relatively small, and the population is not widely representative, all the subjects revealed a decreasing trend in HRV between Scenarios 1 and Scenarios 2. These results are consistent with those observed for different participants and different CW tasks in [16]. Hence, adding more subjects is not expected to affect this result and should be generalizable. The addition of box plots and statistical tests, rather than just individual results, will also make our results more statistically valid and generalizable. A more extensive study with more patients (including BMI, sex, and health status considerations) will be performed in the future. In particular, in clinical populations (e.g., with cardiac pathology), HRV-based CW inference may require additional validation and possibly different analysis techniques.

## 5. Conclusions

We presented a comprehensive study showing that CW alone can drive significant changes in mHRV, as captured by a wearable non-contact MCG sensor. Using our newly designed sensor that is robust against unwanted noise, and utilizing our previously established beat estimation algorithm, we achieved accurate beat detection and mHRV measurements. Our results obtained for n = 10 participants across three scenarios of low and high CW, with and without PA, enabled us to observe a clear decrease in mHRV $Mean_{RR}$ under high CW compared to low CW, regardless of the presence of PA. These findings confirm that, even in the absence of any PA, the human heart’s rhythm reflects the level of cognitive strain. In comparison with prior work and addressing its limitations, we demonstrated improved reliability and understanding of this phenomenon. The conclusion drawn is that mHRV-based metrics, obtainable via unobtrusive MCG technology, are valid indicators of CW. This work advances the state of the art in cognitive monitoring by (i) isolating cognitive effects on a cardiac signal and (ii) providing a path toward practical implementation of wearable MCG as enabled by hardware and algorithmic advancements. In summary, our study establishes a foundation for the future development of smart wearable systems to monitor CW in daily life, with broad implications for personal health, safety, and performance optimization.

## Figures and Tables

**Figure 1 sensors-25-04806-f001:**
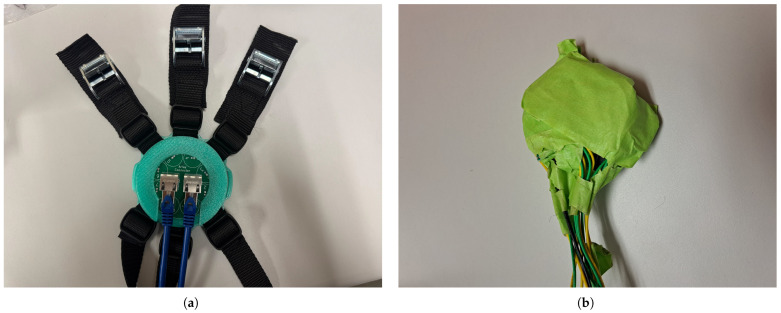
(**a**) Proposed MCG sensor with 3-layer 3D-printed fixture, ethernet cables, and straps. (**b**) Previous MCG sensor with jumper wires kept in place with tape [16].

**Figure 2 sensors-25-04806-f002:**
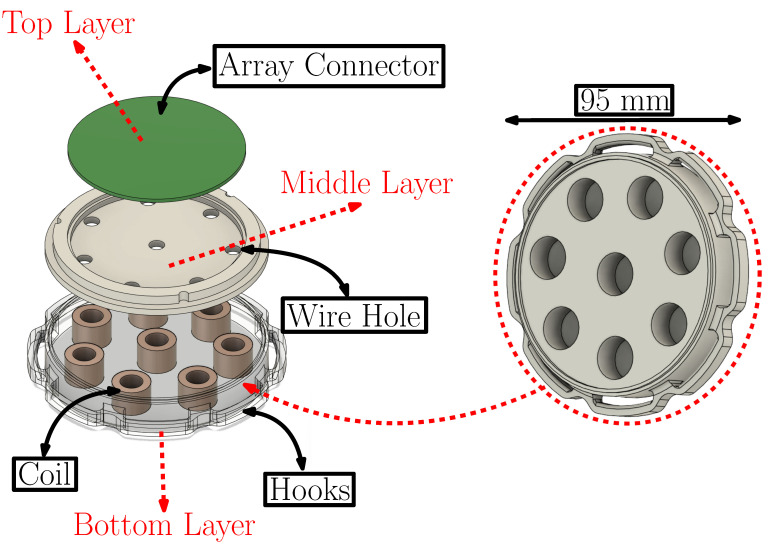
Exploded 3D view of the MCG sensor design.

**Figure 3 sensors-25-04806-f003:**
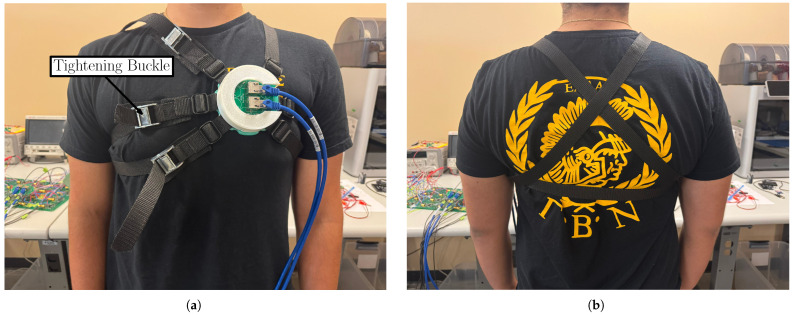
(**a**) Front and (**b**) back view showing how the ratchet straps hold the MCG sensor in place upon a human subject.

**Figure 4 sensors-25-04806-f004:**
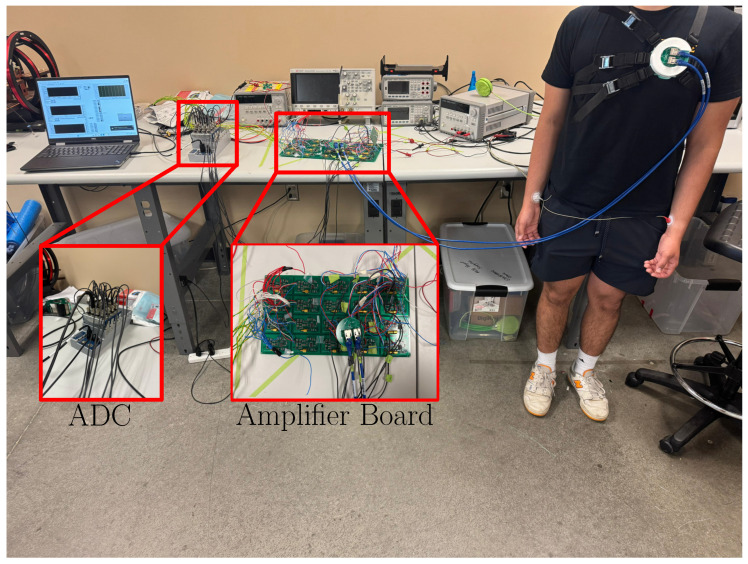
Visualization of the complete MCG sensor setup showing how the ethernet cables connect to the amplifier board, subsequent ADC, and laptop.

**Figure 5 sensors-25-04806-f005:**
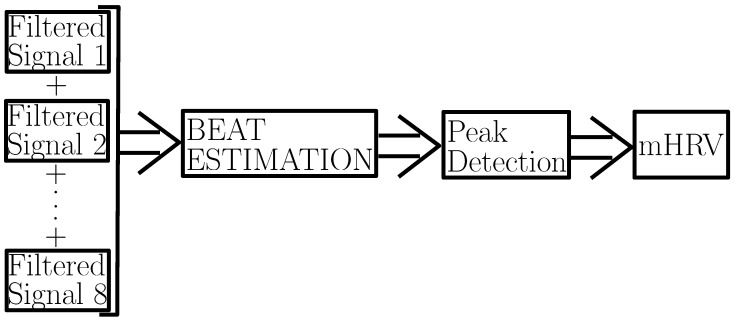
Block diagram of beat estimation.

**Figure 6 sensors-25-04806-f006:**
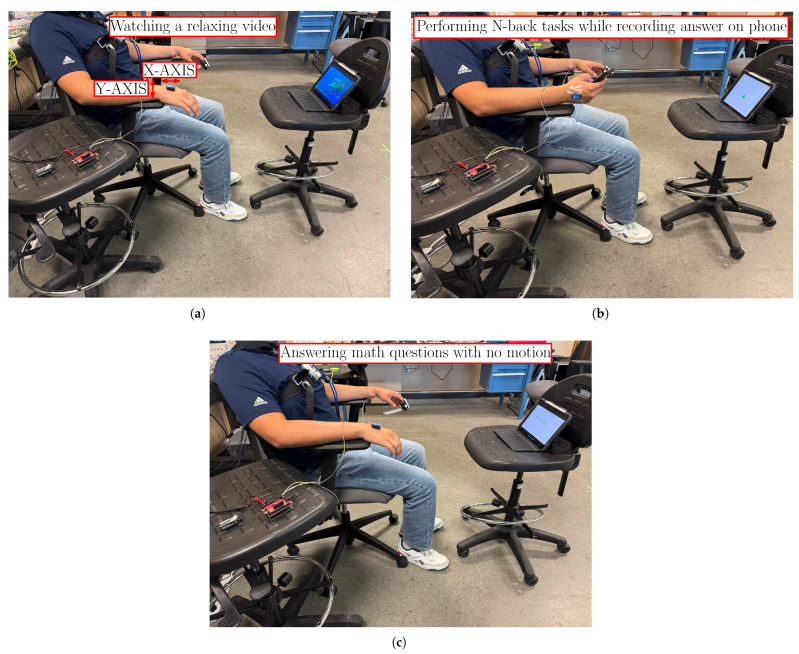
Testing scenarios: (**a**) Scenario 1, (**b**) Scenario 2, and (**c**) Scenario 3.

**Figure 7 sensors-25-04806-f007:**
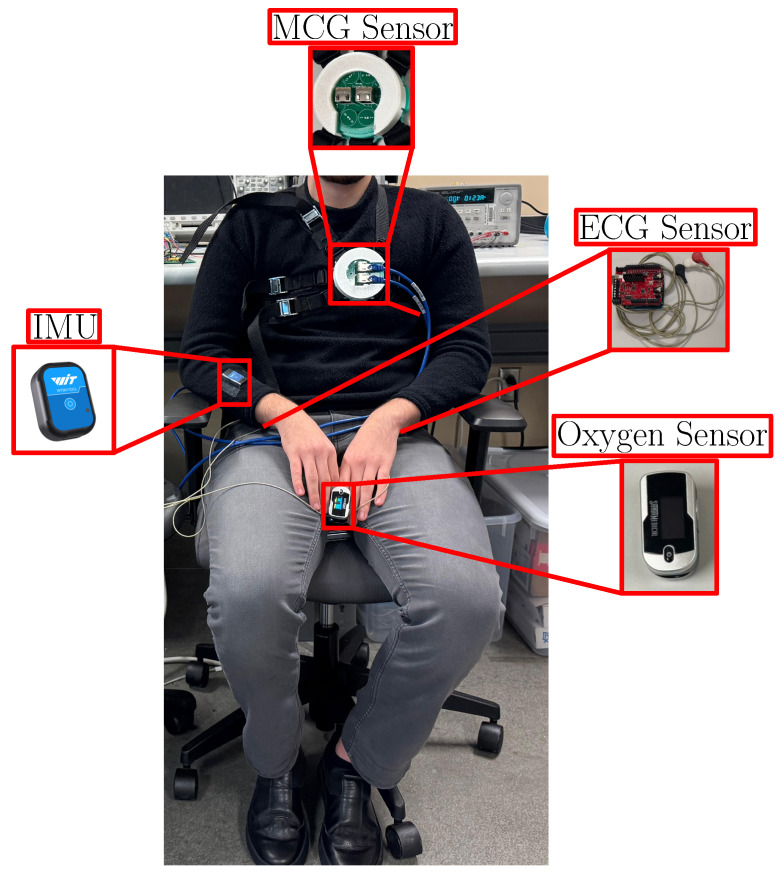
Experimental setup for data collection with visualization on employed sensors.

**Figure 8 sensors-25-04806-f008:**
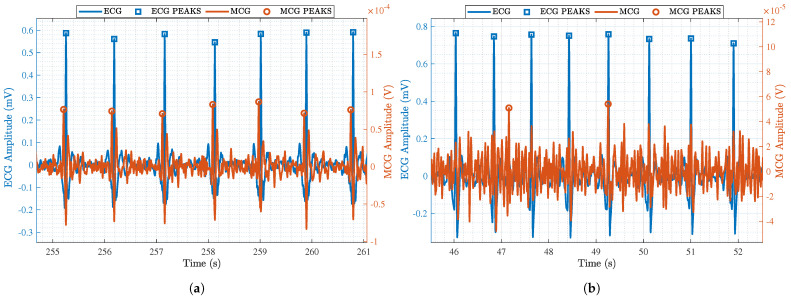
A zoom-in on pre-processed ECG (blue) and MCG (red) data obtained through (**a**) the advanced MCG sensor reported in this paper and (**b**) the MCG sensor reported in [16].

**Figure 9 sensors-25-04806-f009:**
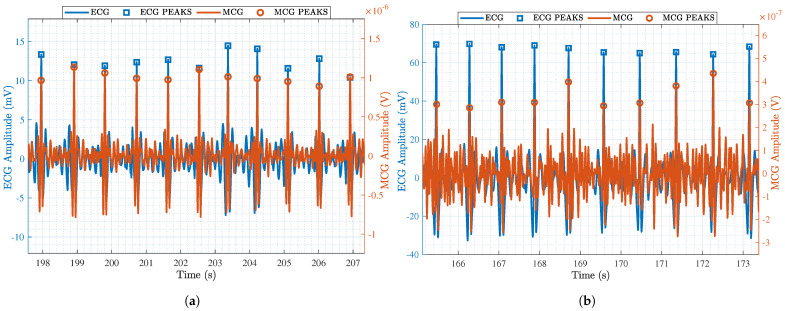
A zoom-in on post-processed ECG (blue) and MCG (red) data obtained through (**a**) the advanced MCG sensor reported in this paper and (**b**) the MCG sensor reported in [16].

**Figure 10 sensors-25-04806-f010:**
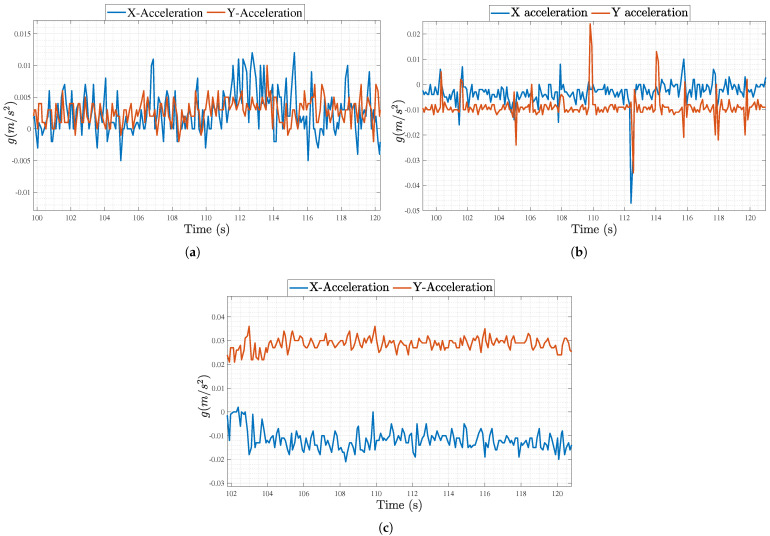
IMU data in X and Y axis in units of *g* (m/s^2^) for (**a**) Scenario 1, (**b**) Scenario 2, and (**c**) Scenario 3.

**Figure 11 sensors-25-04806-f011:**
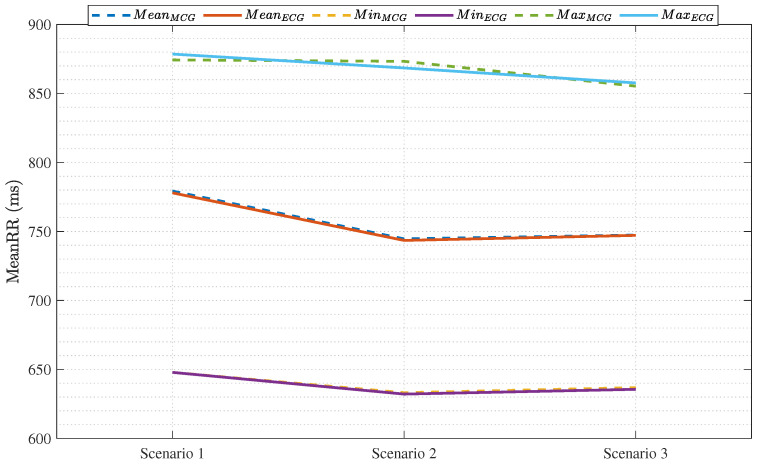
Validation results for MCG: this plot shows how the HRV values obtained through our MCG sensor match those obtained from the ECG sensor.

**Figure 12 sensors-25-04806-f012:**
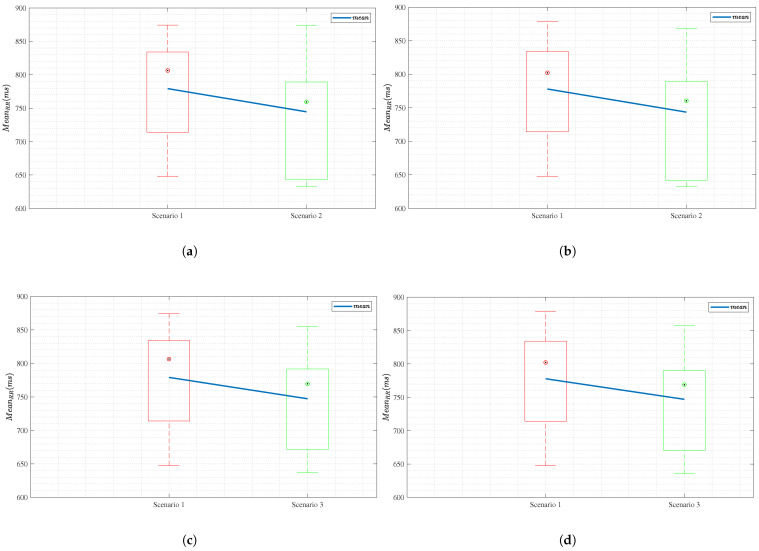
(**a**) Box plot for MCG for Scenarios 1 vs. 2, (**b**) box plot for ECG for Scenarios 1 vs.2, (**c**) box plot for MCG for Scenarios 1 vs. 3, and (**d**) box plot for ECG for Scenarios 1 vs. 3. The blue line that connects the box plots represents the mean of the calculated HRV in each scenario.

**Table 1 sensors-25-04806-t001:** Summary of study participants.

Subject ID	Age	Sex	Height (m)	Weight (Kg)	BMI (kg/m^2^)
Subject 1	23	Male	1.75	60	19.6
Subject 2	23	Male	1.78	85	26.8
Subject 3	33	Male	1.7	71	24.6
Subject 4	20	Male	1.7	76	26.3
Subject 5	25	Male	1.72	67	22.6
Subject 6	23	Male	1.78	72.7	23.0
Subject 7	24	Male	1.82	98	29.5
Subject 8	26	Male	1.82	77.2	24.4
Subject 9	19	Male	1.78	90	28.4
Subject 10	19	Male	1.7	61	21.1

**Table 2 sensors-25-04806-t002:** Summary of detection accuracy.

	Detection Accuracy (%)		
**Subject ID**	**Scenario 1**	**Scenario 2**	**Scenario 3**
Subject 1	96.06	96.67	99.35
Subject 2	100	100	100
Subject 3	96.4	98.61	97.5
Subject 4	97.2	100	99.2
Subject 5	99.6	98.6	100
Subject 6	98.7	90.2	98.3
Subject 7	99.1	99.2	99.7
Subject 8	100	99.6	99.7
Subject 9	96.8	99.6	99.3
Subject 10	99.9	99.5	99.9

## Data Availability

Any inquiries regarding the data presented in this article should be directed to the corresponding author.

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
