# Peer review of "Toward Wearable MagnetoCardioGraphy (MCG) for Cognitive Workload Monitoring: Advancements in Sensor and Study Design"

_sensors, 2025, doi:10.3390/s25154806_

Round 1
Reviewer 1 Report
Comments and Suggestions for Authors
This article proposes a cognitive monitoring system utilizing vibration and acceleration sensors, effectively highlighting the critical role of cognitive workload (CW) regulation in enhancing human performance, reducing errors, and supporting overall well-being. The importance of CW monitoring is well-established, and the limitations of current methods are articulated. The introduction of a novel MCG-based approach as a potential solution is commendable. While the hardware implementation appears robust and the measured data demonstrates sound acquisition, the argumentation could benefit from a more diverse range of examples and a more thorough exploration of potential limitations and practical considerations.
- Specifically, the discussion lacks sufficient depth in exploring the applicability of this approach across various cognitive tasks and real-world scenarios. Further elaboration on the potential challenges and trade-offs associated with this method would strengthen the paper's overall impact. Expanding the illustrative examples to encompass a wider spectrum of cognitive workloads would also enhance the persuasiveness of the proposed solution.
- While the hardware implementation appears promising and the data acquisition seems sound, the argument lacks a diverse range of examples and exploration of potential limitations. It would be beneficial to expand on the practical applications and address potential challenges.
- The paper lacks a comparative analysis with existing methods. Simply monitoring participant responses to a few stimuli doesn't fully address the core problem of cognitive workload that the paper aims to solve. Clarifying the contribution to the field would strengthen the paper's central message.
- The sample size of 10 healthy adults (ranging from adolescence to middle age) may limit the generalizability of the findings. Further discussion on the scope and applicability of this approach to a broader population would be valuable.
- Line 91, p.2, the phrase 'per figure 2' is slightly ambiguous. It would be clearer to specify whether the reader should refer to Figure 2 or if there's an implied connection to the preceding text.
- The separation of heart sounds is a complex task. It would be helpful to discuss whether the MCG sensor may be well-suited to address this challenge and how it might enhance the sensor's capabilities.
- It would be beneficial to clarify the inclusion criteria for participants, specifically addressing whether any participants have pre-existing cardiac conditions (e.g., heart murmurs). Such conditions could potentially affect the interpretation of heart rate variability data, particularly in waveform analysis. Specifying the participant pool would enhance the rigor of the study.
- The experimental scenarios appear somewhat limited, making it difficult to demonstrate a comprehensive comparison of different workload conditions. Expanding the range of tasks and scenarios would strengthen the paper's ability to showcase the advantages of the proposed method.
- I'm curious about the specific design considerations for the MCG sensor. The design, including aspects like the placement and dimensions of the conductive elements (holes, length, size), seems sophisticated. It would be interesting to see a discussion of the rationale behind the optimal design choices.
Author Response
Please find attached the revised and resubmitted version of our manuscript sensors-3762887.
The authors would like to thank the Editor-in-Chief, Associate Editor, and Reviewers for their time and effort spent throughout the review process. Their comments and suggestions proved very valuable in enhancing the quality of the manuscript.
A point-by-point description of changes made to the manuscript in response to the review comments can be found below.
Reviewer 1 Comments and Authors’ Response
This article proposes a cognitive monitoring system utilizing vibration and acceleration sensors, effectively highlighting the critical role of cognitive workload (CW) regulation in enhancing human performance, reducing errors, and supporting overall well-being. The importance of CW monitoring is well-established, and the limitations of current methods are articulated. The introduction of a novel MCG-based approach as a potential solution is commendable. While the hardware implementation appears robust and the measured data demonstrates sound acquisition, the argumentation could benefit from a more diverse range of examples and a more thorough exploration of potential limitations and practical considerations.
Thank you for your comments to help us improve our paper. We have addressed your concerns below and in the revised manuscript.
- Specifically, the discussion lacks sufficient depth in exploring the applicability of this approach across various cognitive tasks and real-world scenarios. Further elaboration on the potential challenges and trade-offs associated with this method would strengthen the paper's overall impact. Expanding the illustrative examples to encompass a wider spectrum of cognitive workloads would also enhance the persuasiveness of the proposed solution.
We have revised Section 4 of the manuscript to include the following discussion: “As is expected, CW can manifest in many contexts beyond the scenarios explored in this pilot study. The selection of the subject scenarios relied on paradigms that have been widely used in the literature to reliably modulate the level of mental effort [1], while the choice of having a controlled laboratory setting arises from the fact that we want to track changes in mHRV solely due to CW. Other memory/cognitive tasks in controlled environments are expected to yield similar results: even if with different intensities, mHRV is expected to drop whenever CW increases. Beyond these scenarios, our MCG approach should, in principle, apply to any task that triggers an autonomic response. For instance, demanding cognitive tasks in safety-critical domains, such as air-traffic control or surgical training, have been shown to tax cognitive resources and mHRV [2]. Likewise, tasks in healthcare (e.g. concussion recovery exercises), education (exam-taking, tutoring), gaming, and immersive training could all benefit from workload monitoring [3][4]. However, applying the proposed approach directly in real-world scenarios would introduce extra variables that might impact mHRV , i.e., mHRV being influenced by several factors beyond CW, such as stress or fitness levels [5]. As such, for this proof-of-concept study, the experimental scenarios were carefully chosen to mimic real-life activities while at the same time limiting external variables that might hinder the mHRV. By contrast, in uncontrolled environments, cognitive load changes might be confounded by other stressors. Any real-world system must account for these.”
- While the hardware implementation appears promising and the data acquisition seems sound, the argument lacks a diverse range of examples and exploration of potential limitations. It would be beneficial to expand on the practical applications and address potential challenges.
We have revised Section 4 of the manuscript to include the following discussion: “In summary, even though the proposed MCG sensor can be used wherever continuous, unobtrusive monitoring of CW is needed, such as applications that include safety monitoring (e.g. warning drowsy or cognitively overloaded drivers/pilots), workplace optimization (e.g., measuring staff workload in manufacturing or control-room environments), and consumer devices (e.g., games or virtual reality systems that adapt to user effort), it is still in the testing stage. It currently requires a snug chest belt and multiple amplifiers, which may not be convenient for all-day wear. Furthermore, it lacks an adequate motion-artifact removal algorithm, making it susceptible to excess motion. Future research should miniaturize and integrate the coils and electronics into a lightweight form factor as well as upgrade the system to address motion artifacts and potentially other confounders.”
- The paper lacks a comparative analysis with existing methods. Simply monitoring participant responses to a few stimuli doesn't fully address the core problem of cognitive workload that the paper aims to solve. Clarifying the contribution to the field would strengthen the paper's central message.
We have revised Section I to provide a more in-depth review of existing methods for monitoring CW.
Specifically, the revised Introduction now states that: “ElectroEncephaloGraphy (EEG), MagnetoEncephaloGraphy (MEG), functional magnetic resonance imaging (fMRI), positron emission tomography (PET), and eye tracking/pupillometry are the most common physiological measures of CW. EEG directly captures the brain’s electrical activity through electrodes positioned on the scalp. This method tracks variations in CW over time and delivers results with high temporal precision [6]. MEG measures the magnetic fields generated by the brain’s electrical activity. It has shown to achieve similar accuracies as EEG [7]. fMRI detects changes in blood flow related to neural activity. It measures the signal that is dependent on blood oxygenation level and has shown to monitor brain activity with high spatial resolution [8] [9]. PET observes metabolic processes in the brain. It can monitor dopamine receptors at rest and while performing a cognitive task [10]. Task-evoked pupillary responses (TEPR) have also shown to effectively estimate the cognitive effort involved in completing a task [11]. However, although the aforementioned techniques have shown to be effective, they can be very complicated to construct and operate, or very expensive to obtain.”
The revised Introduction also states that: “In summary, the key advantage of our work is leveraging MCG as a completely passive cardiac measure for CW. Unlike previous technologies for CW monitoring, our approach requires no skin contact; it senses the heart’s magnetic field directly. This enables continuous monitoring without electrode setup or skin irritation. Furthermore, we operate in normal ambient conditions (no shielding), which is novel. Also, the setup is very cheap to build and operate, as opposed to PET and MEG. Aside from the hardware aspects, another key contribution of this work lies in the separation of changes in HRV from CW and PA. Although literature has confirmed the relationship between changes in CW and HRV, no published work exists that tracks the changes back to CW.”
- The sample size of 10 healthy adults (ranging from adolescence to middle age) may limit the generalizability of the findings. Further discussion on the scope and applicability of this approach to a broader population would be valuable.
We have revised Section 2.4 to clarify that: “For this proof-of-concept study, participants were selected as healthy (i.e., no cardiac conditions) to ensure that changes in mHRV were specifically attributed to CW. This is important because pre-existing cardiac conditions can significantly affect HRV. For example, pathological arrhythmias or heart disease often reduce HRV and alter its pattern [16]. By restricting to healthy adults, we ensured our HRV changes reflected cognitive load rather than underlying pathology. To determine the sample size of human subjects for validation, we based our decision on previous studies of CW monitoring, which included 10[1], 16 [12], 12 [13], 9 [14, 15], and 11[16] participants.”
We have also revised Section 4 with the following discussion: “Though the participant sample size is relatively small and the population is not widely representative, all the subjects revealed a decreasing trend in HRV between Scenarios 1 and Scenarios 2. These results are consistent with those observed for different participants and different CW tasks in [17]. Hence, adding more subjects is not expected to affect this result and should be generalizable. The addition of box plots and statistical tests, rather than just individual results, also make our results more statistically valid and generalizable. A more extensive study with more patients (including BMI, sex, and health status considerations) will be performed in the future. In particular, in clinical populations (e.g. with cardiac pathology), HRV-based CW inference may require additional validation and possibly different analysis techniques.”
- Line 91, p.2, the phrase 'per figure 2' is slightly ambiguous. It would be clearer to specify whether the reader should refer to Figure 2 or if there's an implied connection to the preceding text.
We restructured the statement to refer the reader to both the text and Figure 2: "The following explains in detail the different layers of the sensor, as shown in Figure 2: "
- The separation of heart sounds is a complex task. It would be helpful to discuss whether the MCG sensor may be well-suited to address this challenge and how it might enhance the sensor's capabilities.
We have added the following discussion to Section 2.1: “One possible concern that may arise with the wearable MCG sensor is the possibility of the resulting signal being attributed to heart vibrations or sounds as opposed to MCG. However, as addressed in our previous research [18], the recorded signal is solely attributed to the heart’s magnetic field and the sensor does not measure acoustic heart sounds. Indeed, the coil design has been specifically optimized to pick up MCG signals, while the accompanying signal processing further helps suppress noise while preserving the cardiac signal."
- It would be beneficial to clarify the inclusion criteria for participants, specifically addressing whether any participants have pre-existing cardiac conditions (e.g., heart murmurs). Such conditions could potentially affect the interpretation of heart rate variability data, particularly in waveform analysis. Specifying the participant pool would enhance the rigor of the study.
We have clarified in the revised Section 2.4 that: “Our inclusion criteria (approved by the IRB) required “healthy” status and normal BMI. No screening for subclinical conditions (like undiagnosed arrhythmias) was performed, but none of the volunteers reported any cardiovascular disease or took cardiac medications. By restricting to healthy adults, we ensured our HRV changes reflected cognitive load rather than underlying pathology.”
We have also revised Section 4 to state that: “In particular, in clinical populations (e.g. with cardiac pathology), HRV-based CW inference may require additional validation and possibly different analysis techniques.”
- The experimental scenarios appear somewhat limited, making it difficult to demonstrate a comprehensive comparison of different workload conditions. Expanding the range of tasks and scenarios would strengthen the paper's ability to showcase the advantages of the proposed method.
Please refer to our response to comment #1 raised above.
- I'm curious about the specific design considerations for the MCG sensor. The design, including aspects like the placement and dimensions of the conductive elements (holes, length, size), seems sophisticated. It would be interesting to see a discussion of the rationale behind the optimal design choices.
We have added the following discussion to the revised Section 2.1: “The initial optimization parameters for the coil design were previously established in [19] where the induction coil sensor was designed based on the model of a tightly winded air core coil with an inner diameter Di, outer diameter D, length L, and wire diameter d. In this case, the sensitivity can be written as:
and obtains maximum values if and . Therefore, by selecting a value for D, optimal values for Di and L can be chosen.
As for the array holding these coils, the design specifications were constrained by two main criteria:
- An array that can properly hold the coils in place, and
- A sensor that can be properly fixed on the chest with no jittering
To address the first criterion, we 3D-printed a design with holes that are equivalent to the abovementioned length, L. By doing this, we made sure that the coils are held in place without moving. To address the second criterion, we added hooks and introduced ratchet straps so that the sensor will remain in place.”
References:
[1] Herff, Christian, et al. "Mental workload during n-back task—quantified in the prefrontal cortex using fNIRS." Frontiers in human neuroscience 7 (2014): 935.
[2] Von Janczewski, Nikolai, et al. "A meta-analysis of the n-back task while driving and its effects on cognitive workload." Transportation research part F: traffic psychology and behaviour 76 (2021): 269-285.
[3] Wilbanks, Bryan A., Edwin Aroke, and Katherine M. Dudding. "Using eye tracking for measuring cognitive workload during clinical simulations: literature review and synthesis." CIN: Computers, Informatics, Nursing 39.9 (2021): 499-507.
[4] Wilbanks, Bryan A., and Susan P. McMullan. "A review of measuring the cognitive workload of electronic health records." CIN: Computers, Informatics, Nursing 36.12 (2018): 579-588.
[5] van Ravenswaaij-Arts, Conny MA, et al. "Heart rate variability." Annals of internal medicine 118.6 (1993): 436-447.
[6] Volf, N. V., and A. A. Gluhih. "Background cerebral electrical activity in healthy mental aging." Human Physiology 37.5 (2011): 559-567.
[7] Zhu, Keren, and Asimina Kiourti. "A review of magnetic field emissions from the human body: Sources, sensors, and uses." IEEE Open Journal of Antennas and Propagation 3 (2022): 732-744.
[8] Cappell, Katherine A., Leon Gmeindl, and Patricia A. Reuter-Lorenz. "Age differences in prefontal recruitment during verbal working memory maintenance depend on memory load." Cortex 46.4 (2010): 462-473.
[9] Fischer, Håkan, et al. "Simulating neurocognitive aging: effects of a dopaminergic antagonist on brain activity during working memory." Biological psychiatry 67.6 (2010): 575-580.
[10] Karlsson, Sari, et al. "Modulation of striatal dopamine D1 binding by cognitive processing." Neuroimage 48.2 (2009): 398-404.
[11] Kahneman, Daniel, and Jackson Beatty. "Pupil diameter and load on memory." Science 154.3756 (1966): 1583-1585.
[12] Knoll, Avi, et al. "Measuring cognitive workload with low-cost electroencephalograph." Ifip conference on human-computer interaction. Berlin, Heidelberg: Springer Berlin Heidelberg, 2011.
[13] Hirachan, Niraj, et al. "Measuring cognitive workload using multimodal sensors." 2022 44th annual international conference of the IEEE engineering in medicine & biology society (EMBC). IEEE, 2022.
[14] Mazur, Lukasz M., et al. "Subjective and objective quantification of physician’s workload and performance during radiation therapy planning tasks." Practical radiation oncology 3.4 (2013): e171-e177.
[15] Merkle, Frank, et al. "Evaluation of attention, perception, and stress levels of clinical cardiovascular perfusionists during cardiac operations: a pilot study." Perfusion 34.7 (2019): 544-551.
[16] Schulz, Christian M., et al. "Eye tracking for assessment of workload: a pilot study in an anaesthesia simulator environment." British journal of anaesthesia 106.1 (2011): 44-50.
[17] Wang, Zitong, et al. "Quantifying cognitive workload using a non-contact magnetocardiography (MCG) wearable sensor." Sensors 22.23 (2022): 9115.
[18] Zhu, Keren, and Asimina Kiourti. "Real-time magnetocardiography with passive miniaturized coil array in earth ambient field." Sensors 23.12 (2023): 5567.
[19] Zhu, Keren, et al. "Miniature coil array for passive magnetocardiography in non-shielded environments." IEEE Journal of Electromagnetics, RF and Microwaves in Medicine and Biology 5.2 (2020): 124-131.

Reviewer 2 Report
Comments and Suggestions for Authors
The paper advances the field by significantly improving the hardware and signal processing capabilities of wearable MCG sensors for cognitive workload (CW) monitoring. The BEAT ESTIMATION algorithm and improved sensor fixture are especially notable. Unlike prior work, the authors successfully isolate cognitive workload effects from physical activity (PA), a major methodological improvement that strengthens the validity of the findings. The authors employ a well-controlled study with 10 healthy adults across three test scenarios. The detection accuracy close to 100% is impressive. The comparison with ECG (as a gold standard) adds strong credibility to the reported results, with less than 0.6% average error.This manuscript presents a technically sound, innovative, and well-structured contribution to wearable sensing for cognitive workload monitoring. The improvements over previous MCG systems are clearly demonstrated, and the authors have made significant efforts to validate their approach. However, minor revisions are needed regarding statistical rigor, clarity, and generalizability discussion before it is suitable for publication.
1- All participants are young adult males. While the authors note this is due to sensor garment constraints, it substantially limits generalizability, especially for a sensor intended for broad deployment. authors should Add preliminary data or simulation/phantom-based studies indicating expected performance in female or pediatric populations.
2- While box plots are provided, no formal statistical tests (e.g., ANOVA, paired t-tests) are reported to confirm the significance of HRV differences across scenarios. authors should Include appropriate statistical tests to strengthen the argument that CW changes significantly affect mHRV.
3- The system was tested in controlled laboratory settings. There is no discussion of its performance in semi-realistic or real-world environments (e.g., office, driving simulators). authors should discuss potential signal robustness under mild real-life disturbances or future plans to evaluate this.
4-Some figures (e.g., Figures 8 and 9) could benefit from better axis labeling and clearer legends for improved readability. authors should Add more descriptive figure captions and ensure axis units are clearly stated.
5- The term “MeanRR” is used throughout but should be defined more clearly and early (ideally in the abstract or at first mention). Some formatting issues remain from the submission (e.g., placeholder metadata in the header like “Journal Not Specified,” and inconsistent line numbers) Ensure all metadata is finalized before publication.
6- The authors mention that data is available on request but do not specify any dataset summary or repository. Consider making de-identified data or sample MCG traces publicly available in a repository to support reproducibility.
Author Response
Please find attached the revised and resubmitted version of our manuscript sensors-3762887.
The authors would like to thank the Editor-in-Chief, Associate Editor, and Reviewers for their time and effort spent throughout the review process. Their comments and suggestions proved very valuable in enhancing the quality of the manuscript.
A point-by-point description of changes made to the manuscript in response to the review comments can be found below.
Reviewer 2 Comments and Authors’ Response
The paper advances the field by significantly improving the hardware and signal processing capabilities of wearable MCG sensors for cognitive workload (CW) monitoring. The BEAT ESTIMATION algorithm and improved sensor fixture are especially notable. Unlike prior work, the authors successfully isolate cognitive workload effects from physical activity (PA), a major methodological improvement that strengthens the validity of the findings. The authors employ a well-controlled study with 10 healthy adults across three test scenarios. The detection accuracy close to 100% is impressive. The comparison with ECG (as a gold standard) adds strong credibility to the reported results, with less than 0.6% average error. This manuscript presents a technically sound, innovative, and well-structured contribution to wearable sensing for cognitive workload monitoring. The improvements over previous MCG systems are clearly demonstrated, and the authors have made significant efforts to validate their approach. However, minor revisions are needed regarding statistical rigor, clarity, and generalizability discussion before it is suitable for publication.
Thank you for your comments to help us improve our paper. We have addressed your concerns below and in the revised manuscript.
- All participants are young adult males. While the authors note this is due to sensor garment constraints, it substantially limits generalizability, especially for a sensor intended for broad deployment. Authors should add preliminary data or simulation/phantom-based studies indicating expected performance in female or pediatric populations.
Our previous Section 4 stated that: “Though females were not included in this study, we expect the results obtained in this study to be generalizable. That is, a similar response in mHRV is anticipated with increasing CW, regardless of the participants sex.”
We have now enhanced this discussion to further elaborate on this hypothesis: “Specifically, the basic autonomic mechanisms (i.e., how CW affects HRV) are qualitatively similar across sexes, even if baseline values differ. Known sex differences do exist (for example, women tend to have slightly higher resting heart rate), but no evidence suggests opposite directions of the CW effect. This was shown in several similar works that address CW issues and used only males [1][2], or only females [3] in their testing. This was also demonstrated in our previous work [4], where it was proven that increasing CW will lead to a drop in HRV, among both males and females. Thus, although baseline variations might occur, i.e., variations in the significance of the drop in HRV, HRV is still expected to drop whenever CW increases in both males and females. As for concerns regarding the sensitivity of the sensor due to variations in breast tissue size between males and females, this was proved to have little to not effect during our experiments since our subjects had varying chest size especially since most of them work out”
Future work will test our system in females, but we expect the general pattern, which states that higher workload results in higher HRV, to hold.
- While box plots are provided, no formal statistical tests (e.g., ANOVA, paired t-tests) are reported to confirm the significance of HRV differences across scenarios. Authors should include appropriate statistical tests to strengthen the argument that CW changes significantly affect mHRV.
To add formal statistical tests to strengthen our argument, we performed Welch’s t-tests + Bonferroni. We first define our Hypothesis:
Primary Hypothesis: "Equations are found in the attached document"
We reject the overall null only if both tests are significant at the adjusted significance level (e.g., use Bonferroni correction with α=0.01/2 per test).
After performing the paired t-test, we obtained the following p-values: "Equations are found in the attached document"
At a significance level of 0.01, we reject the null hypothesis and conclude that μ₁ is significantly greater than μ₂ (p < 0.01) and μ₁ is significantly greater than μ3 (p < 0.01).
The following statement was added in the revised Abstract: “A paired Bonferroni t-test with significance α = 0.01 confirms the hypothesis that an increase in CW decreases mHRV.”
We also added the following in the revised Section 3.3:“To confirm the significance of HRV differences across scenarios, we performed a paired Bonferroni t-test with significance α = 0.01 that confirmed our hypothesis. Let μi denote the sample mean of the ith scenario for i ∈ {1, 2, 3}. We obtained a p-value of p(1) = 2.2 × 10−3 for H0(1): μ1 < μ2 and a p-value of p(2) = 4.28 × 10−4 for H0(2): μ1 < μ3.”
- The system was tested in controlled laboratory settings. There is no discussion of its performance in semi-realistic or real-world environments (e.g., office, driving simulators). Authors should discuss potential signal robustness under mild real-life disturbances or future plans to evaluate this.
We have revised Section 4 of the manuscript to include the following discussion: “As is expected, CW can manifest in many contexts beyond the scenarios explored in this pilot study. The selection of the subject scenarios relied on paradigms that have been widely used in the literature to reliably modulate the level of mental effort [5], while the choice of having a controlled laboratory setting arises from the fact that we want to track changes in mHRV solely due to CW. Other memory/cognitive tasks in controlled environments are expected to yield similar results: even if with different intensities, mHRV is expected to drop whenever CW increases. Beyond these scenarios, our MCG approach should, in principle, apply to any task that triggers an autonomic response. For instance, demanding cognitive tasks in safety-critical domains, such as air-traffic control or surgical training, have been shown to tax cognitive resources and mHRV [6]. Likewise, tasks in healthcare (e.g. concussion recovery exercises), education (exam-taking, tutoring), gaming, and immersive training could all benefit from workload monitoring [7][8]. However, applying the proposed approach directly in real-world scenarios would introduce extra variables that might impact mHRV , i.e., mHRV being influenced by several factors beyond CW, such as stress or fitness levels [9]. As such, for this proof-of-concept study, the experimental scenarios were carefully chosen to mimic real-life activities while at the same time limiting external variables that might hinder the mHRV. By contrast, in uncontrolled environments, cognitive load changes might be confounded by other stressors. Any real-world system must account for these.”
We have also revised Section 4 to include the following discussion: “In summary, even though the proposed MCG sensor can be used wherever continuous, unobtrusive monitoring of CW is needed, such as applications that include safety monitoring (e.g. warning drowsy or cognitively overloaded drivers/pilots), workplace optimization (e.g., measuring staff workload in manufacturing or control-room environments), and consumer devices (e.g., games or virtual reality systems that adapt to user effort), it is still in the testing stage. It currently requires a snug chest belt and multiple amplifiers, which may not be convenient for all-day wear. Furthermore, it lacks an adequate motion-artifact removal algorithm, making it susceptible to excess motion. Future research should miniaturize and integrate the coils and electronics into a lightweight form factor as well as upgrade the system to address motion artifacts and potentially other confounders.”
- Some figures (e.g., Figures 8 and 9) could benefit from better axis labeling and clearer legends for improved readability. Authors should add more descriptive figure captions and ensure axis units are clearly stated.
The y-axis of Figures 8 and 9 was changed to emphasize that it denotes the amplitude of ECG and MCG signals. Furthermore, the captions of Figures 8 and 9 were adjusted to clearly describe the plots.
- The term “MeanRR” is used throughout but should be defined more clearly and early (ideally in the abstract or at first mention). Some formatting issues remain from the submission (e.g., placeholder metadata in the header like “Journal Not Specified,” and inconsistent line numbers) Ensure all metadata is finalized before publication.
The following was added to the revised Section 2.2 to further explain MeanRR:
“Here, we calculated mHRV as the mean of the difference in duration between R-peaks, denoted as MeanRR, which was calculated according to: Equations are found in the attached document.
where Ri the index of an R-peak obtained through BEAT ESTIMATION".
As for the formatting issues, we have addressed those that fall within the authors’ responsibility. However, line numbers and headers will be addressed by the journal once the paper is accepted for publication.
- The authors mention that data is available on request but do not specify any dataset summary or repository. Consider making de-identified data or sample MCG traces publicly available in a repository to support reproducibility.
Thank you for noticing this. During the submission process, we noted that the data is not available due to restrictions set forth by the Institutional Review Board (IRB). This statement will be added to the paper by the Journal once the review process is complete.
References:
[1] Safari, MohammadReza, et al. "Classification of mental workload using brain connectivity and machine learning on electroencephalogram data." Scientific Reports 14.1 (2024): 9153.
[2] Martínez-Díaz, Inmaculada C., and Luis Carrasco. "Neurophysiological stress response and mood changes induced by high-intensity interval training: a pilot study." International Journal of Environmental Research and Public Health 18.14 (2021): 7320.
[3] Ismail, Lina, and Waldemar Karwowski. "The brain networks indices associated with the human perception of comfort in static force exertion tasks." Frontiers in Neuroergonomics 6 (2025): 1542393.
[4] Wang, Zitong, et al. "Quantifying cognitive workload using a non-contact magnetocardiography (MCG) wearable sensor." Sensors 22.23 (2022): 9115.
[5] Herff, Christian, et al. "Mental workload during n-back task—quantified in the prefrontal cortex using fNIRS." Frontiers in human neuroscience 7 (2014): 935.
[6] Von Janczewski, Nikolai, et al. "A meta-analysis of the n-back task while driving and its effects on cognitive workload." Transportation research part F: traffic psychology and behaviour 76 (2021): 269-285.
[7] Wilbanks, Bryan A., Edwin Aroke, and Katherine M. Dudding. "Using eye tracking for measuring cognitive workload during clinical simulations: literature review and synthesis." CIN: Computers, Informatics, Nursing 39.9 (2021): 499-507.
[8] Wilbanks, Bryan A., and Susan P. McMullan. "A review of measuring the cognitive workload of electronic health records." CIN: Computers, Informatics, Nursing 36.12 (2018): 579-588.
[9] van Ravenswaaij-Arts, Conny MA, et al. "Heart rate variability." Annals of internal medicine 118.6 (1993): 436-447.

Round 2
Reviewer 1 Report
Comments and Suggestions for Authors
Thank you for addressing many of the issues raised. Here are some minor comments that may help further enhance the quality of the manuscript.
1. The discussion of potential applications is helpful. Could you elaborate on the challenges associated with miniaturizing the sensor and addressing motion artifacts? What specific technologies or signal processing techniques are being considered to overcome these limitations?
2. You state that no screening for subclinical conditions was performed. What steps could be taken to improve the participant screening process to identify individuals who might have underlying cardiac conditions that could affect HRV measurements? For example, would a basic ECG screening be appropriate?
While the added discussion acknowledges the potential for confounding variables in real-world scenarios, it would be beneficial to have a more concrete discussion of how these variables might be addressed.
Author Response
Please find attached the revised and resubmitted version of our manuscript sensors-3762887.
The authors would like to thank the Editor-in-Chief, Associate Editor, and Reviewers for their time and effort spent throughout the review process. Their comments and suggestions proved very valuable in enhancing the quality of the manuscript.
A point-by-point description of changes made to the manuscript in response to the review comments can be found below.
Reviewer 1 Comments and Authors’ Response
The discussion of potential applications is helpful. Could you elaborate on the challenges associated with miniaturizing the sensor and addressing motion artifacts? What specific technologies or signal processing techniques are being considered to overcome these limitations?
We have revised Section 4 of the paper to include the following: "Optimizing the sensor for daily life activities presents several challenges, particularly on the hardware front. While the current amplifier is portable, it is not yet small enough to be considered wearable. Further development is needed to refine and downsize the amplifier so it can match the dimensions of the MCG sensor and be mounted directly on top. Additionally, the Analog-to-Digital Converter (ADC), which digitizes the recorded signals for further processing, should ideally be integrated into the same board as the amplifier. Future research should explore the use of an external electromagnetic transducer designed to suppress common-mode noise. On the algorithmic side, while techniques like adaptive filtering [1], which utilizes data from an Inertial Measurement Unit (IMU), and Independent Component Analysis (ICA) [2] are being considered, incorporating an IMU alone isn’t sufficient. It is equally important to understand the origins of these motion artifacts in order to model and effectively eliminate them."
You state that no screening for subclinical conditions was performed. What steps could be taken to improve the participant screening process to identify individuals who might have underlying cardiac conditions that could affect HRV measurements? For example, would a basic ECG screening be appropriate?
We have revised Section 2.4 of the paper to include the following: "For future studies involving participants with known or suspected cardiac conditions, we can leverage the partnership with Nationwide Children's Hospital to recruit patients who have already been appropriately screened. This approach helps ensure that our findings are not confounded by underlying cardiac issues. Additionally, we can use a 3-lead ECG system to aid in detecting arrhythmias, as different types of cardiac irregularities are known to alter the morphology of the heartbeat [3][4]."
References:
[1] Ram, M. Raghu, et al. "A novel approach for motion artifact reduction in PPG signals based on AS-LMS adaptive filter." IEEE Transactions on Instrumentation and Measurement 61.5 (2011): 1445-1457.
[2] Kim, Byung S., and Sun K. Yoo. "Motion artifact reduction in photoplethysmography using independent component analysis." IEEE transactions on biomedical engineering 53.3 (2006): 566-568.
[3] Couderc, Jean-Philippe. "Measurement and regulation of cardiac ventricular repolarization: from the QT interval to repolarization morphology." Philosophical Transactions of the Royal Society A: Mathematical, Physical and Engineering Sciences 367.1892 (2009): 1283-1299.
[4] Xue, Joel, et al. "Electrocardiographic morphology changes with different type of repolarization dispersions." Journal of electrocardiology 43.6 (2010): 553-559.
